# Additional outreach effort of providing an opportunity to obtain a kit for fecal immunochemical test during the general health check-up to improve colorectal cancer screening rate in Japan: A longitudinal study

**Misuzu Fujita** *, Takehiko Fujisawa, Akira Hata

Department of Health Research, Chiba Foundation for Heath Promotion and Disease Prevention, Chiba City, Chiba, Japan

* mi-hujita@keko-chiba.or.jp

**Data Availability Statement:** The minimal anonymized dataset are available form the Zendo

## Abstract

### Objectives

A sufficient screening rate is indispensable to optimize the positive impact of colorectal cancer (CRC) screening. This study aimed to evaluate the effect of an additional outreach of providing an opportunity to obtain a kit for fecal immunochemical test (FIT) during the general health check-up to increase CRC screening rate.

### Methods

This was a longitudinal study using pre-existing data in Kujukuri Town, Japan. The town provided CRC screening in the fiscal year (FY) 2017 using an existing procedure for all beneficiaries of the National Health Insurance, whereas in FY 2018, an additional outreach effort was made to only those with an even number of age (exposed group), who were offered an opportunity to obtain a kit for FIT at the time of general health check-ups but not to those with an odd number of age (control group). To estimate the effectiveness, generalized estimating equation (GEE) with individuals as clusters was performed.

### Results

In total, 3,530 individuals were included (1,708 in the control group and 1,822 in the exposed group). GEE showed significant interaction between the groups (control and exposed) and FYs (2017 and 2018) (p<0.001), indicating that the change in CRC screening rate from 2017 to 2018 was significantly different between the two groups. Although an achieved actual rate of 17.1% in the exposed group in FY 2018 was low, the additional outreach increased the rate by 5.8 percentage point (95% confidence interval, 3.5–8.1) compared with an existing rate.

database (Zenodo, DOI: 10.5281/zenodo. 3951839).

**Funding:** The author(s) received no specific funding for this work.

**Competing interests:** The authors have declared that no competing interests exist.

## Conclusions

Additional outreach of providing an opportunity to obtain a kit for FIT at the time of the general health check-up improved the CRC screening rate. However, screening rate achieved by this strategy remained low, indicating further efforts is required.

## Introduction

The incidence of colorectal cancer (CRC) globally increased by 38% between 2007 and 2017 from 1.3 million to 1.8 million, and an estimated 896,000 died from the disease in 2017 [1]. In Japan, CRC is the most common malignancy and the second leading cause of cancer-related death, with 158,127 incident cases [2] and 50,681 deaths [3] in 2017.

CRC screening using fecal blood tests (FBTs) including a guaiac-based fecal occult blood test and a fecal immunochemical test (FIT) has been proven to effectively reduce CRC-related mortality [4–6]. A point of great importance in the CRC screening would be a sufficient screening rate to maximize the benefit. Accordingly, in 2014, the American Cancer Society's National Colorectal Cancer Roundtable set a national goal of screening rate to be 80% by 2018 [7]. Achieving the goal could translate into 203,000 potential lives saved from 2013 through 2030 [8]. In Japan, even though an accurate CRC screening rate is unavailable owing to the lack of registration system, some statistics have been published. According to a Report on Regional Public Health Services and Health Promotion Services, the CRC screening rate in the population aged 40–69 years was only 8.4% in 2017 [9]. According to Comprehensive Survey of Living Conditions, based on a self-reported questionnaire, the CRC screening rate in 2016 was 41.4% [10]. In either case, the screening rate in Japan remained at a much lower level than 50% of the government's target value. Thus, effective strategies to improve CRC screening rate are keenly anticipated.

Previous studies have evaluated the effectiveness of several strategies to improve CRC screening rate, such as mailed FBT outreach [11–13], advanced notices and follow-up reminders to the patient [14–16], tailored navigation intervention [16], and financial incentives to individuals [17]. Another strategy to improve the rate is visit-based FBT outreach, in which FBT kits are provided to the patients when they visit a site, such as a clinic or health center, to avail other health care services. For example, visit-based FBT outreach at a clinic site for influenza vaccinations has increased the CRC screening rate [18, 19]. Although various efforts to increase CRC screening rate have been reported, the rate is still unsatisfactory, especially in Japan.

Kujukuri town provided a kit for FIT to the beneficiaries of National Health Insurance (NHI) during the health check-up program launched by the Japanese government with the aim to improve the CRC screening rate as a practice. Under the health check-up program, insurers are obliged to provide the check-up to all beneficiaries aged 40–74 years. Municipalities are the insurers of NHI for self-employed workers, farmers, retirees, and the unemployed, and they have to provide general health check-up to beneficiaries. Therefore, all beneficiaries of NHI had an opportunity to undergo the general health check-up provided by the municipalities. Furthermore, because municipalities are main providers of CRC screening in Japan, they could implement this outreach for beneficiaries of NHI.

This study aimed to evaluate the effectiveness of the outreach effort, which is performed as a practice, to increase the CRC screening rate. Toward this goal, we performed a longitudinal study using real world data.

## Materials and methods

### CRC screening in Japan

The Guideline for Colorectal Cancer Screening in Japan strongly recommends CRC screening using FBTs, preferably FIT, as a population-based screening of individuals aged 40 years or older. Based on the recommendation, CRC screening is usually provided by municipalities or workplaces in Japan.

### General health check-up program targeting metabolic syndrome

In April 2008, the Japanese government launched a general health check-up program targeting metabolic syndrome, in which insurers were obliged to provide the check-up to all beneficiaries aged between 40 and 74 years [20]. Japan has three major health care insurance systems: a health insurance for government and company employees; the NHI administrated by municipalities for self-employed workers, farmers, retirees, and the unemployed; and a health insurance for elderly adults aged 75 years or older [21]. Because municipalities are insurers of NHI, they should provide the general health check-up to beneficiaries. Additionally, they are recommended to provide a CRC screening to them as mentioned above. In fact, Kujukuri town offered the general health check-up and CRC screening to beneficiaries in both FY 2017 and 2018.

### Subjects and study setting

Subjects in this study were beneficiaries of NHI aged between 40 and 74 years in Kujukuri Town, Chiba Prefecture, Japan. The town is a rural and seaside area located about 64 km southeast from Tokyo and had a population of 16,510 in 2015. The town is an area with aging population, in which the residents over the age of 65 years is 35.1%, which is higher than that in whole of Japan (26.6%). CRC screening rate of this town was lower than that of national average (5.7% vs. 8.4%) in 2017, according to a Report on Regional Public Health Services and Health Promotion Services [9]. CRC screening procedures in FY 2017 and 2018 are shown in Fig 1. In 2017, CRC screening was performed using an existing screening procedure to all beneficiaries. In FY 2018, to improve the CRC screening rate, outreach methods were provided to those who had an even number of age at the end of FY 2018. This facility was not offered to those who had odd number of age to avoid confusion and overburden the health center. For them, CRC screening was performed using the existing procedure in FY 2018 as well FY 2017. Accordingly, we defined the exposed group as the beneficiaries with an even number of age at the end of FY 2018 and the control group as those with an odd of number age. To evaluate the effectiveness of this outreach, CRC screening rate in FY2017 and 2018 were compared between the groups. Analyzed subjects were beneficiaries of NHI who belonged to the NHI from April 1, 2017 to March 31, 2019 (FY 2017 and 2018).

In 2017, colorectal cancer screening performed using an existing screening procedure to all beneficiaries. In 2018, an additional outreach to improve CRC screening rate performed for only beneficiaries with an even number of age at the end of fiscal year 2018.

We defined the beneficiaries who brought the kits to the health care center as those who underwent screening (gray shading).

### Existing procedure in the town

Beneficiaries who were willing to undergo a CRC screening were required to go to the sites for gastric cancer screening to obtain a kit for FIT. After collecting a fecal sample at home, they were advised to return with it to the health care center in Kujukuri town. This information on

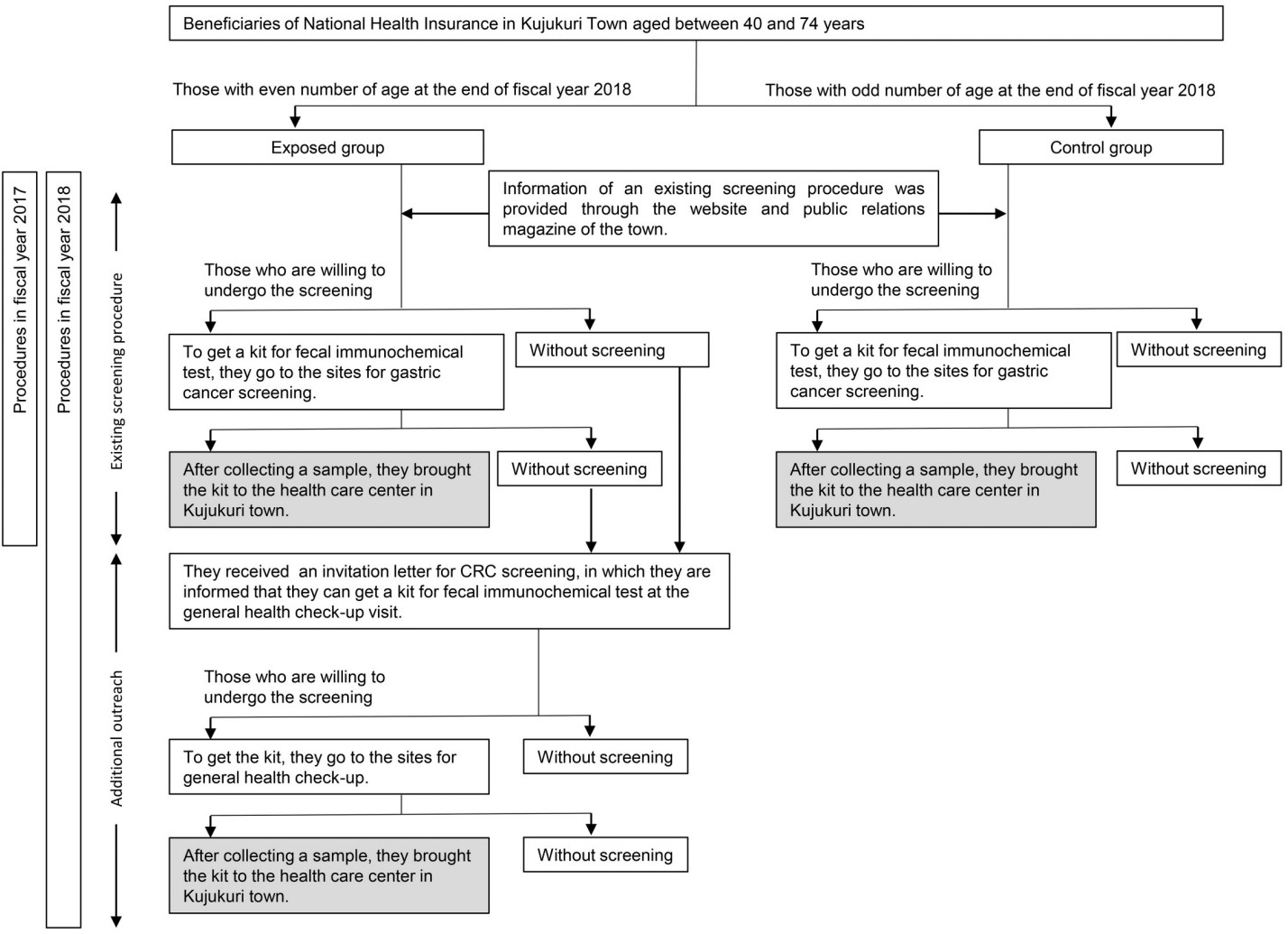

**Fig 1. Schematic diagram representing the process for conducting colorectal cancer screening in fiscal year 2017 and 2018.**

existing procedure for CRC screening was shared through the website and public relations magazine of the town.

## Additional outreach to improve CRC screening rate

Those beneficiaries who had not received a kit for FIT by the existing procedure, an invitation letter for CRC screening was mailed along with that for the general health check-up to them, informing them of the availability of the kit for FIT that they can obtain on their health check-up visit. Those, who were willing to undergo a CRC screening were given a kit for FIT in exchange for an invitation letter at the general health check-up site.

After collecting the fecal sample at home, they brought it to the health care center in Kuju-kuri town. This additional procedure allowed the subjects to obtain the kit at the time of the general screening visiting.

The out-of-pocket payment for CRC screening was 400 Japanese yen, equivalent to 3.68 US dollars at an exchange rate of 1 US dollar = 108.71 Japanese yen on December 11, 2019.

## Statistical analyses

To estimate the effectiveness of the outreach, we performed generalized estimating equation (GEE) which is often used to analyze longitudinal data [22]. We set individuals as clusters to incorporate similarities within individuals into the model, set up the distribution to binominal, and link function to logit. The dependent variable was whether the subjects received CRC screening or not (received or not received). We defined the beneficiaries who brought the kit to the health care center as those who received the screening as shown in gray shade in Fig 1. The independent variables were group (control or exposed) and FY (2017 or 2018). Interaction term between group and FY was entered in the model. Sex and age were also entered in the model as covariates for adjustment. Age was categorized into four groups (40–49, 50–59, 60–69, and 70–74 years) because the association between age and CRC screening rate might not be linear. Sample size was not preset, because this is a study to evaluate the outreach performed as a practice using existing data acquired by Kujukuri town. All statistical analyses were performed using STATA software version 15.0 (STATA LP, College Station, TX). A two-tailed *p*-value of <0.05 was considered significant.

## Ethics statement

This study was approved by the Research Ethics Committee of Chiba Foundation for Health Promotion and Disease Prevention (approval number: R1-4). The study was conducted in accordance with the Declaration of Helsinki and the Ethical Guidelines for Medical and Health Research Involving Human Subjects. The additional outreach to improve CRC screening as mentioned above was not planned for the study. This outreach was provided to residents as a usual health service. We utilized the existing data collected by Kujukuri town to evaluate the effectiveness of the outreach. Because this was a retrospective study using existing data, the need for informed consent was waived. All personal information (e.g., names and telephone numbers) was completely removed from the dataset for this study and a correspondence table to individuals was not made. The procedure for anonymization was performed in the town before the dataset was provided to us.

## Results

Number of beneficiaries of Kujukuri town NHI from April 1, 2017 to March 31, 2019 (FY 2017 and 2018) was 3,530. Data on these subjects were used in this study (1,708 in the control group and 1,822 in the exposed group). The subject characteristics are shown in Table 1. There was a 1-year difference in the mean age between the two groups, but sex and CRC screening rate in FY 2017 were similar. The result of GEE is shown in Table 2. Notably, there was a significant interaction between the groups and FYs (p <0.001), indicating that the change in CRC screening rate from 2017 to 2018 was significantly different between the two groups. To facilitate interpretation of this association, the screening rate estimated by this model is shown in Fig 2. The CRC screening rate in 2017 was similar between the two groups; however, gradient of the rate from 2017 to 2018 was apparently steeper in the exposed group than in the control group. The CRC screening rate significantly increased from 2017 to 2018 in both the exposed group and the control group by 9.0 percentage point (95% confidence interval [CI], 7.5–10.5) and 3.5 percentage point (95% CI, 2.3–4.7), respectively. The difference in the screening rate in 2018 between the two groups, which was calculated by subtracting the rate in the control group from that in the exposed group, was estimated to be 5.8 percentage point (95% CI, 3.5–8.1). This result indicated that although the achieved actual rate of 17.1% in the exposed group in FY 2018 was low, the additional outreach can increase the CRC screening rate by 5.8 percentage point.

**Table 1. Subject characteristics.**

|  | Control group | Exposed group |
|---|---|---|
| **Number** | 1,708 | 1,822 |
| **Sex (men)[1]** | 888 (0.520) | 978 (0.537) |
| **Age at the end of fiscal year 2017[2]** | 62.5 (9.1) | 63.6 (9.2) |
| **40–49 years[1]** | 252 (0.15) | 267 (0.15) |
| **50–59 years[1]** | 273 (0.16) | 274 (0.15) |
| **60–69 years[1]** | 787 (0.46) | 843 (0.46) |
| **70–74 years[1]** | 396 (0.23) | 438 (0.24) |
| **Subjects who underwent screening in fiscal year 2017[1]** | 143 (0.084) | 157 (0.086) |
| **Subjects who underwent screening in fiscal year 2018[1]** | 205 (0.120) | 326 (0.179) |

Control group: NHI beneficiaries in Kujukuri town with odd number of age at the end of fiscal year 2018.

Exposed group: NHI beneficiaries in Kujukuri town with even number of age at the end of fiscal year 2018.

[1] Number (proportion) shown.

[2] Mean (standard deviation) shown.

## Discussion

This study evaluated the effectiveness of an additional outreach providing an opportunity for insurance beneficiaries in the exposed group to obtain a kit for FIT during the general health check-up to improve the CRC screening rate. We found that the outreach significantly increased the CRC screening rate by 5.8 percentage point. This strategy has several advantages. First, it can be implemented in all municipalities in Japan because the general health check-up targeting metabolic syndrome is performed nationwide in Japan. Second, it may be widely used even if the number of health care staff is limited because it does not require attentive implementation in contrast to other strategies such as one-on-one education. Third, the

**Table 2. Results of the generalized estimating equation to estimate colorectal cancer screening rate.**

| Variables | Odds ratio | 95% confidence interval | p-value |
|---|---|---|---|
| **Group** | | | |
| Control group | 1.00 | | |
| Exposed group | 1.04 | 0.82–1.32 | 0.747 |
| **Fiscal year** | | | |
| Fiscal year 2017 | 1.00 | | |
| Fiscal year 2018 | 1.50 | 1.30–1.73 | <0.001 |
| **Group × fiscal year** | 1.56 | 1.27–1.92 | <0.001 |
| **Sex** | | | |
| Men | 1.00 | | |
| Women | 1.71 | 1.42–2.05 | <0.001 |
| **Age (years)** | | | |
| 40–49 | 1.00 | | |
| 50–59 | 1.61 | 1.08–2.40 | 0.020 |
| 60–69 | 2.52 | 1.80–3.52 | <0.001 |
| 70–74 | 2.43 | 1.70–3.48 | <0.001 |

Control group: NHI beneficiaries in Kujukuri town with odd number of age at the end of fiscal year 2018.

Exposed group: NHI beneficiaries in Kujukuri town with even number of age at the end of fiscal year 2018.

We defined individuals as clusters and set up the distribution to binominal and link function to logit.

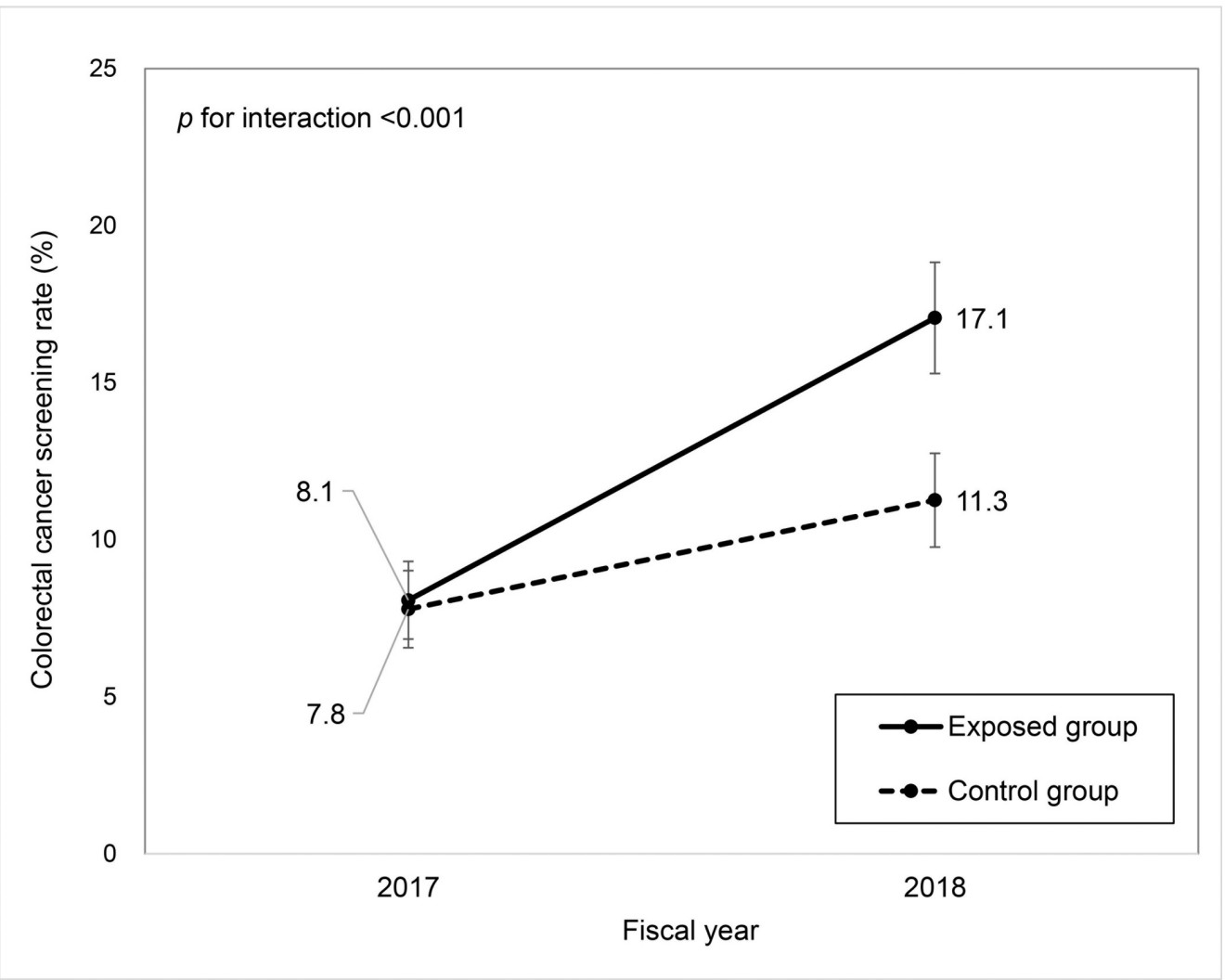

**Fig 2. Colorectal cancer screening rate estimated using a generalized estimating equation model.** Control group: NHI beneficiaries in Kujukuri town with odd number of age at the end of fiscal year 2018. Exposed group: NHI beneficiaries in Kujukuri town with even number of age at the end of fiscal year 2018. For calculation of the colorectal cancer screening rates, the proportion of sex and age is set as follows: the proportion of women is 47.1% and the proportions of 40–49, 50–59, 60–69, and 70–74 years were 14.7%, 15.5%, 46.2%, and 23.6%, respectively, which are obtained from the distribution in our subjects. The difference in cancer screening rate between the two groups in 2018 is estimated to be 5.8 percentage point (95% CI, 3.5–8.1). The difference in cancer screening rate between fiscal year 2017 and fiscal year 2018 in the control group is estimated to be 3.5 percentage point (95% CI, 2.3–4.7). The difference in cancer screening rate between fiscal year 2017 and fiscal year 2018 in the exposed group is estimated to be 9.0 percentage point (95% CI, 7.5–10.5).

number of kits for FIT can be conserved because it is provided only to the subjects who are willing to receive a CRC screening. However, although this strategy significantly increased the CRC screening rate and has certain advantages, its effectiveness was limited. The screening rate in the exposed group in FY 2018 was only 17.1%. One reason for this might be the large effort required from the subjects. The subjects who are willing to undergo CRC screening have to go to the site of the general health check-up to obtain the kit for FIT and then have to go to the health care center to submit their kit after sampling. The strategies to overcome this short-coming could be adoption of other municipalities' approaches in improving CRC screening in Japan. For example, Kobe city accepts the requests for FIT kits by telephone and Kyoto city by

mail. Upon the request, the cities send the kit to the residents by mail and they return samples to the cities by mail. These strategies do not require the subjects to visit a site such as a health center. Additionally, the three strengths of the outreach we claimed are also adaptable to these strategies. However, the effectiveness has not been evaluated as far as we know. Thus, we have no idea on how much CRC screening rate increased with these strategies. We believe efforts to evaluate the strategies and make the results public like the current study will contribute to improved CRC screening rate.

Previous study evaluated several strategies to improve CRC screening rate [11–17]. Of those, direct mailed FIT outreach, in which kits for FIT were sent directly to the subjects by mail showed the greatest effectiveness [11–13]. A systematic review reported that direct mailed FIT outreach increased the screening rate to a median of 21.5 percentage point [23], and a meta-analysis indicated that the strategy can increase the screening rate by 2.28 times [24]. The efficacy of the direct mailed outreach is likely to be apparently greater than that of the outreach evaluated in this study. Unfortunately, no study has evaluated the effectiveness of direct mailed FIT outreach in Japan, and most of the municipalities currently do not implement this strategy because of concerns on wasted FIT kits and financial limitations. To further improve the CRC screening rate, the effectiveness of other strategies should be evaluated, and the effective strategies identified should be combined.

In our study, the CRC screening rate significantly increased not only in the exposed group, but also in the control group. One possible reason for this would be a ripple effect of this outreach. When the subjects in the control group wanted to undergo CRC screening at the time of the general health check-up, we accepted the request. If their family members or neighbors belonged to the exposed group, their motivation to receive the screening might be promoted. Considering the ripple effect, the effectiveness of our strategy would be underestimated. Additionally, the increased screening rate in the control group might have been affected by other factors that changed over time. For example, laws, campaigns, and advertisements could change public opinion and individual consciousness for CRC. However, these factors cannot be determined in general. If we only had the exposed group, we cannot distinguish the effect of the outreach and the effect of the factors that changed over time. The inclusion of a control group allowed us to estimate the effectiveness of the outreach by comparing the screening rate between the two groups.

This study has several limitations. First, group assignment was not randomized; the subjects were instead categorized by age. However, the effect of selection bias would be small because the assignment depended on only age but not on intentions of the subjects and researchers. Indeed, sex and CRC screening rate in FY 2017 were similar between the two groups. Second, the generalizability of our findings is limited because we only included residents from one town in Japan. The effectiveness of this outreach might depend on the characteristics of the municipality, such as existing procedure for CRC screening, baseline CRC screening rate, and characteristics of the residents. Third, there is no registration system for CRC screening in Japan, and the true CRC screening rate could not be determined. However, because our subjects were beneficiaries of the NHI, which is an insurance for self-employed workers, farmers, retirees, and the unemployed, they would have limited opportunity to receive the screening provided by sectors other than the municipalities.

In conclusion, this longitudinal study using real world data revealed that additional outreach providing an opportunity to obtain a kit for FIT during the general health check-up improved the CRC screening rate by 5.8 percentage point in Japan. This strategy is advantageous because it can be applied in all municipalities. However, the strategy alone could not adequately improve the CRC screening rate to an acceptable level, and thus, further efforts are needed to increase the CRC screening rate in Japan.

## Acknowledgments

We greatly appreciate the valuable help and support of the staff members of Kujukuri town hall. We would also like to thank Editage (www.editage.jp) for English language editing.

## Author Contributions

**Conceptualization:** Misuzu Fujita, Akira Hata.

**Data curation:** Misuzu Fujita.

**Formal analysis:** Misuzu Fujita.

**Investigation:** Misuzu Fujita.

**Methodology:** Misuzu Fujita, Akira Hata.

**Project administration:** Takehiko Fujisawa, Akira Hata.

**Resources:** Misuzu Fujita, Takehiko Fujisawa, Akira Hata.

**Supervision:** Takehiko Fujisawa, Akira Hata.

**Validation:** Misuzu Fujita.

**Visualization:** Misuzu Fujita.

**Writing – original draft:** Misuzu Fujita.

**Writing – review & editing:** Takehiko Fujisawa, Akira Hata.

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
