## [Decision Letter · Decision Letter 0]

7 Jul 2020

PONE-D-20-15630

Additional outreach effort of providing an opportunity to obtain a kit for fecal immunochemical test during the general health check-up to improve colorectal cancer screening rate in Japan: A quasi-randomized control study

PLOS ONE

Dear Dr. Fujita,

Thank you for submitting your manuscript to PLOS ONE. After careful consideration, we feel that it has merit but does not fully meet PLOS ONE’s publication criteria as it currently stands. Therefore, we invite you to submit a revised version of the manuscript that addresses the points raised during the review process.

We look forward to receiving your revised manuscript.

Kind regards,

Toshiyuki Ojima, M.D., Ph.D

Academic Editor

PLOS ONE

Additional Editor Comments:

It would not be appropriate to define the design of this study as a quasi-RCT.

The authors evaluated the measures that the town had implemented as a practice, not for the research.

It seems that there was no research protocol in advance, no registrations of RCT, no criteria for inclusion/exclusion for residents, and intention to analysis was not done.

I believe that studies using real world data, such as this study, would make a significant contribution to public health. However, the study design should be defined as a longitudinal study, not a quasi-RCT.

Journal Requirements:

Reviewers' comments:

Reviewer's Responses to Questions

**Comments to the Author**

1. Is the manuscript technically sound, and do the data support the conclusions?

Reviewer #1: Yes

Reviewer #2: Yes

2. Has the statistical analysis been performed appropriately and rigorously? 

Reviewer #1: Yes

Reviewer #2: Yes

3. Have the authors made all data underlying the findings in their manuscript fully available?

Reviewer #1: No

Reviewer #2: No

4. Is the manuscript presented in an intelligible fashion and written in standard English?

Reviewer #1: Yes

Reviewer #2: Yes

5. Review Comments to the Author

Reviewer #1: Thank you for your great work.

I enjoyed reading the article.

My comments as below are based on the STROBE checklist for observational study (http://www.equator-network.org/reporting-guidelines/strobe/)

and I hope these comments improve your article.

Major concerns

#1. I would like to know why the authors regarded individuals as clusters. Please describe the rationale to define individuals as clusters in detail.

#2. Please explain how to decide the participants. Were sum of control group (1,708) and intervention group (1,822) all beneficiaries of National Health Insurance in the town aged between 40 and 74 years? In other words, Was the study exhaustive survey?

#3. Please let readers know how to handle people who moved in or out during the study period.

Reviewer #2: General comments

This study evaluated the outreach strategy at the general health check-up to increase CRC screening rate. The authors found that additional outreach of providing an opportunity to obtain a kit for FIT at the health check-up site increased the CRC screening rate compared with usual screening procedure. The strategy used would be feasible in the real world and the results are scientifically meaningful.

I offer the following comments and suggestions.

1. I would recommend to add the description, especially in the Methods section, according to each item of the CONSORT 2010 Statement, which is an evidence-based, minimum set of recommendations for reporting randomized trials.

2. Page 4, Lines 41-48.

Please add the description about the scientific background and rationale to select the outreach strategy at health-check up among several other strategies to increase CRC screening rate in the Introduction.

3. Page 5, Lines 74-

Please add the description about the characteristics of the targeted town, such as population density, aging rates, and cancer mortality rates compared to the national average.

4. Page13, Lines 189-194.

One of the major barriers to this strategy seems to be that beneficiaries need to visit the health center twice. In some municipalities, an application form for CRC screening is also enclosed in the health-check up ticket, and the test kit is sent to residents who return the application. With this method, residents will not have to visit the health center again. The three strengths of the strategy described in the Discussion would apply for this method as well. It would be great if authors could describe explicitly about the strengths of the strategy used in this study, including a comparison with the method in some municipalities.

5. Showing the process outcome as well as the primary outcome is essential to improve the implementation of interventions. In this study, the proportion of people in each flow in Figure1 will be significant to assess the which phase is a barrier to take a CRC screening. Please fill in the number and proportion in each flow in both groups in Figure1 (e.g., number of proportions of those who went to the sites to get a kit and those who brought the kit to health center, by existing procedure and additional outreach, respectively) and add the results and interpretation for the process outcome as well.

6. Same as other countries, Japan has socio-economic disparities in cancer screening rates. Do you have any data about the characteristics of people with which the current strategy has improved the CRC screening rate? If there was a disparity in the CRC screening rate by socio-economic, sex, or area at the time of 2017 (baseline), it would be great if authors could additionally examine whether that disparity has been narrowed or widened by this outreach strategy.

6. PLOS authors have the option to publish the peer review history of their article (what does this mean?). If published, this will include your full peer review and any attached files.

Reviewer #1: **Yes: **Makoto Kaneko

Reviewer #2: No

---

## [Author Response · Author response to Decision Letter 0]

3 Aug 2020

Responses to the Editor’s and Reviewers’ Comments

Manuscript reference number: PONE-D-20-15630

We appreciate the editor’s and reviewers’ comments toward improving our paper. We have revised the manuscript to address all the comments received, and our point-by-point responses are provided below.

Academic Editor

Comment

It would not be appropriate to define the design of this study as a quasi-RCT.

The authors evaluated the measures that the town had implemented as a practice, not for the research.

It seems that there was no research protocol in advance, no registrations of RCT, no criteria for inclusion/exclusion for residents, and intention to analysis was not done.

I believe that studies using real world data, such as this study, would make a significant contribution to public health. However, the study design should be defined as a longitudinal study, not a quasi-RCT.

Response

We agree with your indication. We have changed the description of “a quasi-randomized control study” to “a longitudinal study” in the Title, Abstract, Introduction, and Discussion sections. Additionally, the word, “intervention” was changed to “exposed” as appropriate.

Reviewer #1

Major comments

Comment 1

I would like to know why the authors regarded individuals as clusters. Please describe the rationale to define individuals as clusters in detail.

Response

The statistical method we used, generalized estimating equation (GEE), is often applied to analyze longitudinal and other correlated response data. In our study, the response was whether individual received CRC screening or not, which was repeatedly measured at fiscal year 2017 and 2018. For such data, it is obvious that the responses from the same individual tend to be “more alike,” thus incorporating within-subject and between-subject variations into model fitting is necessary. To incorporate similarities within individuals into the model, we set individuals as clusters.

We added the explanation in the Materials and Methods section as follows:

“To estimate the effectiveness of the outreach, we performed generalized estimating equation (GEE) which is often used to analyze longitudinal data [22]. We set individuals as clusters to incorporate similarities within individuals into the model, set up the distribution to binominal, and link function to logit.” (Page 8, Lines 119-122).

Comment 2 and 3

Please explain how to decide the participants. Were sum of control group (1,708) and intervention group (1,822) all beneficiaries of National Health Insurance in the town aged between 40 and 74 years? In other words, Was the study exhaustive survey?

Please let readers know how to handle people who moved in or out during the study period.

Response

Yes, our study was exhaustive survey.

We already added the description of the study subjects in the Statistical analyses section, however, the reviewer’s comment prompted us to move the description to the Subjects and study setting section for better understanding. The descriptions are as follows:

“To evaluate the effectiveness of this outreach, CRC screening rate in FY2017 and 2018 were compared between the groups. Analyzed subjects were beneficiaries of NHI who belonged to the NHI from April 1, 2017 to March 31, 2019 (FY 2017 and 2018).” (Page 6, Lines 89-92)

Additionally, explanation about the subjects was added in the Results section as follows.

“Number of beneficiaries of Kujukuri town NHI from April 1, 2017 to March 31, 2019 (FY 2017 and 2018) was 3,530. Data on these subjects were used in this study (1,708 in the control group and 1,822 in the exposed group).” (Page 9, Lines 145-147)

Reviewer #2

Comment 1

I would recommend to add the description, especially in the Methods section, according to each item of the CONSORT 2010 Statement, which is an evidence-based, minimum set of recommendations for reporting randomized trials.

Response

Following the academic editor's suggestion, we have modified the study design from quasi-RCT to a longitudinal study. We checked STROBE statement, in accordance with this statement, we added a description about sample size as follows:

“Sample size was not preset, because this is a study to evaluate the outreach performed as a practice using existing data acquired by Kujukuri town.” (Page 8, Lines 128-129)

Comment 2

Page 4, Lines 41-48.

Please add the description about the scientific background and rationale to select the outreach strategy at health-check up among several other strategies to increase CRC screening rate in the Introduction.

Response

The outreach strategy in our study was not planned for research. It was implemented as a practice by the town. Thus, we could not select the outreach strategy.

We have revised the Introduction section as follows:

“Although various efforts to increase CRC screening rate have been reported, the rate is still unsatisfactory, especially in Japan.

Kujukuri town provided a kit for FIT to the beneficiaries of National Health Insurance (NHI) during the health check-up program launched by the Japanese government with the aim to improve the CRC screening rate as a practice. Under the health check-up program, insurers are obliged to provide the check-up to all beneficiaries aged 40–74 years. Municipalities are the insurers of NHI for self-employed workers, farmers, retirees, and the unemployed, and they have to provide general health check-up to beneficiaries. Therefore, all beneficiaries of NHI had an opportunity to undergo the general health check-up provided by the municipalities. Furthermore, because municipalities are main providers of CRC screening in Japan, they could implement this outreach for beneficiaries of NHI.

This study aimed to evaluate the effectiveness of the outreach effort, which is performed as a practice, to increase the CRC screening rate. Toward this goal, we performed a longitudinal study using real world data.” (From Page 4, Line 47 to Page 5, Line 59)

Comment 3

Page 5, Lines 74-

Please add the description about the characteristics of the targeted town, such as population density, aging rates, and cancer mortality rates compared to the national average.

Response

Several characteristics of the study town in the Materials and Methods section are as follows:

“Subjects in this study were beneficiaries of NHI aged between 40 and 74 years in Kujukuri Town, Chiba Prefecture, Japan. The town is a rural and seaside area located about 64 km southeast from Tokyo and had a population of 16,510 in 2015. The town is an area with aging population, in which the residents over the age of 65 years is 35.1%, which is higher than that in whole of Japan (26.6%). CRC screening rate of this town was lower than that of national average (5.7% vs. 8.4%) in 2017, according to a Report on Regional Public Health Services and Health Promotion Services [9].” (Page 6, Lines 77-82)

Calculated population density of the town is higher than the national average (675 people / km2 vs. 336 people / km2), presumably due to large inhabitable forest area in Japan as a whole. The town’s population density is not higher compared to those of neighboring municipalities, special ward of Tokyo (14,776 people / km2) and Chiba city (3,576 people / km2). Thus, we decided not to describe the number to avoid misleading information.

In addition, accurate cancer mortality rate of the town was regrettably unavailable. 

Comment 4

Page13, Lines 189-194.

One of the major barriers to this strategy seems to be that beneficiaries need to visit the health center twice. In some municipalities, an application form for CRC screening is also enclosed in the health-check up ticket, and the test kit is sent to residents who return the application. With this method, residents will not have to visit the health center again. The three strengths of the strategy described in the Discussion would apply for this method as well. It would be great if authors could describe explicitly about the strengths of the strategy used in this study, including a comparison with the method in some municipalities.

Response

Thank you for your suggestion. We totally agree.

We have added some explanations about the strategy implemented in some municipalities in the Discussion section as follows:

“The strategies to overcome this shortcoming could be adoption of other municipalities’ approaches in improving CRC screening in Japan. For example, Kobe city accepts the requests for FIT kits by telephone and Kyoto city by mail. Upon the request, the cities send the kit to the residents by mail and they return samples to the cities by mail. These strategies do not require the subjects to visit a site such as a health center. Additionally, the three strengths of the outreach we claimed are also adaptable to these strategies. However, the effectiveness has not been evaluated as far as we know. Thus, we have no idea on how much CRC screening rate increased with these strategies. We believe efforts to evaluate the strategies and make the results public like the current study will contribute to improved CRC screening rate.” (Page 14, Lines 209-217)

Comment 5

Showing the process outcome as well as the primary outcome is essential to improve the implementation of interventions. In this study, the proportion of people in each flow in Figure 1 will be significant to assess the which phase is a barrier to take a CRC screening. Please fill in the number and proportion in each flow in both groups in Figure1 (e.g., number of proportions of those who went to the sites to get a kit and those who brought the kit to health center, by existing procedure and additional outreach, respectively) and add the results and interpretation for the process outcome as well.

Response

We agree with your suggestion. However, only limited information to evaluate main outcome were provided by the town. We regret that we are unable to respond to this comment adequately.

Comment 6

Same as other countries, Japan has socio-economic disparities in cancer screening rates. Do you have any data about the characteristics of people with which the current strategy has improved the CRC screening rate? If there was a disparity in the CRC screening rate by socio-economic, sex, or area at the time of 2017 (baseline), it would be great if authors could additionally examine whether that disparity has been narrowed or widened by this outreach strategy.

Response

We agree the indication is an interesting perspective. However, the items related to socio-economic, such as income and educational background, were not obtained from the town. We could clarify the impact of this strategy on socio-economic disparity if the data are available.

For reference, the subjects were limited to the exposed group (the subjects with even number of age), and the interaction between year and sex and between year and age were evaluated. Both interactions were not statistically significant, indicating that the disparities by sex and age were neither narrowed nor widened by the strategy. Although we think that the disparity is an important issue, this result was not added in the manuscript because it is not the main purpose of this study.

---

## [Decision Letter · Decision Letter 1]

18 Aug 2020

Additional outreach effort of providing an opportunity to obtain a kit for fecal immunochemical test during the general health check-up to improve colorectal cancer screening rate in Japan: A longitudinal study

PONE-D-20-15630R1

Dear Dr. Fujita,

We’re pleased to inform you that your manuscript has been judged scientifically suitable for publication and will be formally accepted for publication once it meets all outstanding technical requirements.

Kind regards,

Toshiyuki Ojima, M.D., Ph.D

Academic Editor

PLOS ONE

Additional Editor Comments (optional):

Reviewers' comments:

Reviewer's Responses to Questions

**Comments to the Author**

1. If the authors have adequately addressed your comments raised in a previous round of review and you feel that this manuscript is now acceptable for publication, you may indicate that here to bypass the “Comments to the Author” section, enter your conflict of interest statement in the “Confidential to Editor” section, and submit your "Accept" recommendation.

Reviewer #1: All comments have been addressed

Reviewer #2: All comments have been addressed

2. Is the manuscript technically sound, and do the data support the conclusions?

Reviewer #1: Yes

Reviewer #2: Yes

3. Has the statistical analysis been performed appropriately and rigorously? 

Reviewer #1: Yes

Reviewer #2: Yes

4. Have the authors made all data underlying the findings in their manuscript fully available?

Reviewer #1: Yes

Reviewer #2: No

5. Is the manuscript presented in an intelligible fashion and written in standard English?

Reviewer #1: Yes

Reviewer #2: Yes

6. Review Comments to the Author

Reviewer #1: (No Response)

Reviewer #2: The authors have addressed all comments sufficiently. The methods are much clearer now and discussions are more insightful.

7. PLOS authors have the option to publish the peer review history of their article (what does this mean?). If published, this will include your full peer review and any attached files.

Reviewer #1: No

Reviewer #2: No

---

## [Editor Report · Acceptance letter]

21 Aug 2020

PONE-D-20-15630R1 

Additional outreach effort of providing an opportunity to obtain a kit for fecal immunochemical test during the general health check-up to improve colorectal cancer screening rate in Japan: A longitudinal study 

Dear Dr. Fujita:

I'm pleased to inform you that your manuscript has been deemed suitable for publication in PLOS ONE. Congratulations! Your manuscript is now with our production department. 

Kind regards, 

on behalf of

Dr. Toshiyuki Ojima 

Academic Editor

PLOS ONE